# Topology-Aware Dynamic Reweighting for Distribution Shifts on Graph

Weihuang Zheng [* 1]   Jiashuo Liu [* 2]   Jiaxing Li [1]   Jiayun Wu [2]   Peng Cui [2]   Youyong Kong [1]

## Abstract

Graph Neural Networks (GNNs) are widely used for node classification tasks but often fail to generalize when training and test nodes come from different distributions, limiting their practicality. To address this challenge, recent approaches have adopted invariant learning and sample reweighting techniques from the out-of-distribution (OOD) generalization field. However, invariant learning-based methods face difficulties when applied to graph data, as they rely on the impractical assumption of obtaining real environment labels and strict invariance, which may not hold in real-world graph structures. Moreover, current sample reweighting methods tend to overlook topological information, potentially leading to suboptimal results. In this work, we introduce the Topology-Aware Dynamic Reweighting (TAR) framework to address distribution shifts by leveraging the inherent graph structure. TAR dynamically adjusts sample weights through gradient flow on the graph edges during training. Instead of relying on strict invariance assumptions, we theoretically prove that our method is able to provide distributional robustness, thereby enhancing the out-of-distribution generalization performance on graph data. Our framework's superiority is demonstrated through standard testing on extensive node classification OOD datasets, exhibiting marked improvements over existing methods.

## 1. Introduction

Node classification tasks have widespread applications in real life, such as advertising recommendation (Jiang et al., 2023), social network anomaly detection (Tang et al., 2022), and more. Recently, Graph Neural Networks (GNNs) have become cornerstone models for node classification. However, these GNN models typically assume that the training and test graph data are drawn from the same distribution, which does not always hold in practice. In real-world graph data, sample selection bias (Fan et al., 2022; He et al., 2020) and the methods used for graph construction (Qiao et al., 2018; Zhou et al., 2023) often brings distribution shifts between training nodes and test nodes. For instance, In WebKB (Pei et al., 2020) datasets, web pages (nodes) and categories (labels) are heavily affected by the university they originate from, leading to distribution shifts among nodes drawn from different universities. Therefore, addressing distribution shifts is crucial to improving the practical effectiveness of GNNs for node classification.

To address the distribution shift problem on the node classification task, recent works (Wu et al., 2021; Xia et al., 2024; Wu et al., 2022; Liu et al., 2023) adopt the idea of invariant learning. Invariant learning (Arjovsky et al., 2019; Liu et al., 2021a) stems from the causal inference literature, and now becomes one of the key approaches to solve distribution shifts problem on graphs. The core concept of invariant learning is to identify invariant features with stable prediction mechanisms across different environments (distributions), thereby mitigating performance degradation under distribution shifts. Despite its success, methods based on invariant learning are built upon strong invariance assumptions that lack further validation for their actual validity (Liu et al., 2024). In different environments, invariant features may not exist, as the environment can influence the mapping relationship between features and labels. For example, in social networks, users with the same profile (age, interests, etc.) may exhibit different behaviors depending on their location. In region A (where personalized ads are allowed), they may frequently click on ads, while in region B (where privacy regulations limit ad tracking), their click-through rate may significantly decrease.

Besides, sample reweighting methods, particularly those based on distributionally robust optimization (DRO), have also been used to address distribution shifts in graph data (Hu & Hong, 2013; Wang et al., 2024; Gui et al., 2022). These methods reweight the probability distribution of training samples within a predefined distribution set to identify the worst-case distribution under which the model performs the poorest. By optimizing against this worst-case distribu-

[*]Equal contribution [1]School of Computer Science and Engineering, Southeast University [2]Department of Computer Science and Technology, Tsinghua University. Correspondence to: Youyong Kong <kongyouyong@seu.edu.cn>.

*Proceedings of the 42nd International Conference on Machine Learning*, Vancouver, Canada. PMLR 267, 2025. Copyright 2025 by the author(s).

tion, these methods theoretically ensure improved robustness to potential distribution shifts. For example, KL-DRO (Hu & Hong, 2013) improves robustness by assigning higher sample weights to high-risk samples. DR-GNN (Wang et al., 2024) builds on the idea of KL-DRO to address distribution shift issues in recommender systems. Furthermore, (Gui et al., 2022) demonstrated the effectiveness of Group DRO (Sagawa et al., 2019) for node classification under distribution shifts by assigning greater losses to the group of nodes with the highest loss in the graph. However, these methods directly adopt general-domain reweighting strategies without considering the topological structure of graphs, which may lead to suboptimal results.

In this work, we focus on the problem of distributional shift in node classification tasks. Instead of adopting invariant learning that relies on strict invariance assumptions, as in most current approaches, we adopt a sample reweighting method. Specifically, we propose the **T**opology-**A**ware Dynamic **R**eweighting (**TAR**) framework to remedy the issues above. During training, TAR assigns more densities to high-risk nodes, thereby prompting the prediction model to prioritize these nodes. Additionally, we theoretically prove that this method can identify the local worst distribution, and by optimizing this distribution, the model can achieve improved robustness. Our main contributions are as follows:

- **Method:** Our TAR framework involves a minimax procedure, where the inner maximization problem learns sample probability densities (also referred to as sample weights) under the entropy and topology constraints, and the outer minimization problem optimizes the GNN model under the learned distribution. For the reweighting scheme (inner problem), as illustrated in Figure 1, we perform gradient flow along the graph edges to assign greater sample weight to nodes with higher loss. In this way, we *incorporate the topological structure information* into the learning of sample probability densities. Furthermore, we propose leveraging graph extrapolation to expand the given data distribution, further enhancing the model's resilience to potential distribution shifts.

- **Theory:** In Section 3.2, we theoretically prove that our gradient flow procedure is equivalent to finding the local worst-case distribution, which enhances the distributional robustness of our GNN model. We also characterize the error rate introduced by our gradient flow as $e^{-CT_{\text{in}}}$ ($T_{\text{in}}$ is the number of steps).

- **Experiments:** Experimental results on standard OOD node classification datasets demonstrate the effectiveness of the TAR framework, showing its superiority in addressing distributional shift problems.

## 2. Preliminaries

**Notations.** $X \in \mathcal{X}$ denotes the covariates, $Y \in \mathcal{Y}$ denotes the target, $\mathbb{P}_s(X, Y)$ and $\mathbb{P}_t(X, Y)$ represent the joint source distribution and the target distribution, abbreviated with $\mathbb{P}_s$ and $\mathbb{P}_t$ respectively. The prediction model is denoted by $f_\theta(\cdot) : \mathcal{X} \rightarrow \mathcal{Y}$, for which we use graph neural networks (GNN) throughout this paper. $[N] = \{1, 2, \ldots, N\}$ denotes the set of integers from 1 to $N$. A weighted finite graph is denoted by $G_0 = (V, E, W)$, where $V = \{v_1, \ldots, v_N\}$ is the node set, $E$ is the edge set, and $W = (w_{ij})_{(i,j) \in E}$ are the edge weights. $\mathcal{N}(i)$ denotes the set of adjacent nodes for the $i$-th node.

**Problem setting.** In this paper, we focus on the problem of distribution shift in node classification. Specifically, our goal is to learn a predictor $f_\theta^*$ that generalizes well under target distribution. Formally, we aim to find $f_\theta^*$ that satisfy:

$$f_\theta^* = \arg\min_{f_\theta} \mathbb{E}_{(X,Y)\sim\mathbb{P}_t} \left[ \ell(f_\theta(X), Y) \right],$$

where $\ell(\cdot, \cdot)$ is a predefined loss function. As the target distribution is not available during training, previous approaches typically optimize the empirical loss on the training source distribution $\mathbb{E}_{(X,Y)\sim\mathbb{P}_s} \left[ \ell(f_\theta(X), Y) \right]$ as a surrogate. However, the distribution shifts between $\mathbb{P}_s$ and $\mathbb{P}_t$ can lead to the failure of the predictor.

**Sample Reweighting.** Formally, sample reweighting is formulated as:

$$f_\theta^* = \arg\min_{f_\theta} \max_{\mathbb{P} \in S(\mathbb{P}_s)} \mathbb{E}_{(X,Y)\sim\mathbb{P}} \left[ \ell(f_\theta(X), Y) \right],$$

where $S(\mathbb{P}_s)$ indicates the potential distribution set around the source distribution. By adjusting the probability densities $q(x, y)$ (i.e. reweight samples) to simulate the worst potential shifts in the inner maximization, this method can ensure the predictor's robustness when encountering potential unseen distributions. Typically, $S(\mathbb{P})$ is constrained by $d(\mathbb{P}, \mathbb{P}_s) \leq \hat{d}$, where $d(\cdot, \cdot)$ is a distance metric measuring the difference between two distributions, $\hat{d}$ is a predefined constant. This constraint ensures that the reweighting avoids unrealistic distributions, focusing only on plausible shifts.

**Discrete geometric Wasserstein distance.** We review some key concepts and introduce the discrete geometric Wasserstein distance (Chow et al., 2017), where we adopt the notations used in (Chow et al., 2017; Liu et al., 2022).

The (empirical) probability set supported on all nodes of $G_0$ is denoted as:

$$\mathcal{P}(G_0) = \left\{ p = (p_i)_{i=1}^N : \sum_{i=1}^N p_i = 1, p_i \geq 0, \text{for } i \in [N] \right\},$$

which contains all empirical distributions on the node set $V$, and the interior of $\mathcal{P}(G_0)$ is denoted as $\mathcal{P}_o(G_0)$. A

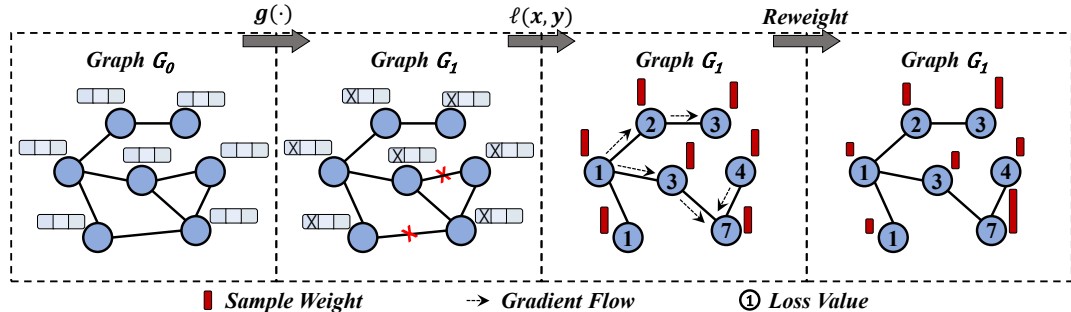

*Figure 1.* Illustration of TAR. Given graph $g_0$, we first conduct graph extrapolation to generate $G_1$. Next, we compute the classification loss, initially assigning equal sample weights to each node. Subsequently, we perform topology-aware reweighting. Specifically, nodes with higher loss values attract more sample weight through gradient flow from neighboring nodes. Finally, nodes with higher losses are assigned greater weights, while nodes with identical loss values exhibit different weights due to variations in their topological positions.

*velocity* field $v = (v_{ij})_{i,j \in V} \in \mathbb{R}^{N \times N}$ on graph $G_0$ is a skew-symmetric matrix on the edge set $E$:

$$v_{ij} = \begin{cases} -v_{ij} & \text{if } (i,j) \in E, \\ 0 & \text{otherwise.} \end{cases}$$

Given the probability function $p \in \mathcal{P}(G_0)$ and a velocity field $v$, the *flux* function is defined as the product $pv \in \mathbb{R}^{N \times N}$:

$$pv := (v_{ij}\xi_{ij}(p))_{(i,j) \in E},$$

where $\xi_{ij}(p)$ is a predefined "cross-sectional area", typically interpolated with the associated nodes' densities $p_i, p_j$. To ensure the positiveness of $p$ during optimization, we adopt the upwind interpolation from statistical mechanics (Hsu, 1981): $\xi_{ij}(p) = \mathbb{I}(v_{ij} > 0)p_j + \mathbb{I}(v_{ij} \leq 0)p_i$ throughout this paper, which relies on the corresponding velocity field. Intuitively, this characterizes the "flux" of sample density from node $i$ to $j$. Based on this, the *divergence* vector of $pv$ on graph $G_0$ is defined as:

$$\text{div}_{G_0}(pv) := -\left( \sum_{j \in V:(i,j) \in E} \sqrt{w_{ij}} v_{ij}\xi_{ij}(p) \right)_{i=1}^N \in \mathbb{R}^N,$$

which is supposed to lie in the tangent space of $P_o(G_0)$. Intuitively, the $i$-th element in $\text{div}_{G_0}(pv)$ sums over all the in-fluxes and out-fluxes along edges to a certain target node $i$, with each source edge $j$ transporting a probability density $\sqrt{w_{ij}} v_{ij}\xi_{ij}(p)$.

Now we define the discrete geometric Wasserstein distance:

**Definition 2.1** (Discrete Geometric Wasserstein Distance). Given a finite graph $G_0$, for any pair of distributions $p^0, p^1 \in P_o(G_0)$, the discrete geometric Wasserstein distance is defined as:

$$\mathcal{GW}_{G_0}^2(p^0, p^1) := \inf_v \left\{ \int_0^1 \frac{1}{2} \sum_{(i,j) \in E} \xi_{ij}(p(t)) v_{ij}^2 dt \right\},$$

s.t. $\dfrac{dp}{dt} + \text{div}_{G_0}(pv) = 0, p(0) = p^0, p(1) = p^1,$

where the infimum is taken over all velocity fields on $G_0$, and $\xi_{ij}(p)$ is a pre-defined interpolation function between $p_i$ and $p_j$. Note that $p(t)$ is a continuously differentiable curve $p(t) : [0,1] \to P_o(G_0)$, which characterizes the probability densities at time $t$.

*Remark* 2.2. In contrast with the conventional Wasserstein distance defined within Euclidean space, the geometric Wasserstein distance necessitates that the transportation of probability density is along the geodesic determined by the graph structure $G_0$. In particular, the constraint $\frac{dp}{dt} + \text{div}_{G_0}(pv) = 0$ imposes the condition that the change in probability density remains continuous with respect to $G_0$.

## 3. Method

Motivated by the discrete geometric Wasserstein distance in Definition 2.1, we propose the Topology-Aware Dynamic Reweighting (TAR) algorithm to deal with node classification tasks under distribution shifts. As shown in Figure 1, our proposed TAR consists of two key stages. In the first stage, we employ graph extrapolation to expand the original data distribution, thereby generating a more diverse and comprehensive set of node samples. Building upon this expanded graph data, the second stage introduces a topology-aware dynamic sample reweighting mechanism, which aims to simulate the potential worst-case distribution. By training the model to perform well on this constructed distribution, our method enhances the model's robustness against unseen distribution shifts.

Consider source data $D_s = \{(x_i, y_i)\}_{i=1}^N$ and the corresponding graph structure $G_0 = (V, E, W)$. Denote the empirical marginal distribution as $\hat{\mathbb{P}}_s$, the overall objective

of our TAR algorithm is formulated as:

$$\min_{\theta \in \Theta} \max_{q \in \mathcal{P}_o(G_0)} \underbrace{\sum_{i=1}^{N} q_i \ell(f_\theta(x_i), y_i)}_{\text{Weighted loss}}$$

$$\underbrace{- \beta \cdot \sum_{i=1}^{N} q_i \log q_i}_{\text{Entropy penalty}} \underbrace{- \lambda \cdot \mathcal{GW}_{G_0}^2(\hat{\mathbb{P}}_s, q)}_{\text{Topology penalty}}, \quad (1)$$

where $\beta$ is the hyper-parameter, and the objective function in general is a minimax optimization over model parameters $\theta$ and sample probability densities $q$. Note that for the parameter $\lambda$, we set it as $\lambda = \frac{1}{2\tau}$ in our optimization (for details, please refer to Section 3.1). During training, the inner maximization assigns more densities to high-risk samples, thereby prompting the prediction model to prioritize these points. This approach aims for a uniformly robust performance across all samples on the graph and helps mitigate potential distribution shifts. Moreover, to mitigate the risk of overemphasizing unrealistic distributions (e.g., noisy nodes accumulating excessive densities), we introduce entropy and topology penalties as regularization terms. These penalties integrate topology information for smooth sample weight assignments along the graph structure.

**Illustrations.** Here we make some remarks on our objective function:
(a) *Entropy penalty*: $(-\sum_{i=1}^{N} q_i \log q_i)$ represents the entropy of empirical probability distribution $q$. As illustrated in Section 3.2, this term serves as a non-linear graph Laplacian operator that encourages sample weights to be smooth along the manifold, avoiding extreme sample weights in the weighted distribution.
(b) *Topology penalty*: $\mathcal{GW}_{G_0}^2(\hat{\mathbb{P}}_s, q)$ represents the optimal transport distance between the source distribution $\hat{\mathbb{P}}_s$ and the weighted distribution $q$, measured along the graph structure. This term explicitly integrates topology information to enforce minimal changes in sample densities along the manifold. As detailed in Section 3.1, this term transfers the optimization of sample densities from Euclidean space to geometric Wasserstein space. Here, densities are constrained to change exclusively along the graph structure. This enforcement encourages *local smoothness* of sample densities relative to the manifold, which helps to mitigate against potential noisy samples and edges.

### 3.1. Topology-Aware Dynamic Reweighting

The main challenge of Problem 1 lies in the computation of discrete geometric Wasserstein distance $\mathcal{GW}_{G_0}^2(\hat{\mathbb{P}}_s, q)$, which itself involves an complicated optimization problem and does not have an analytical form. In this section, we propose to leverage Wasserstein gradient flow to approximately solve the inner maximization problem. The whole algorithm

**Algorithm 1** Topology-Aware Dynamic Reweighting (TAR) Scheme

---

**Input:** Labeled training nodes $D = \{(x_i, y_i)\}_{i=1}^{N}$, learning rate $\gamma$, gradient flow iterations $T_{\text{in}}$, entropy term $\beta$, graph structure $G_0 = (V, E, W)$.
**Initialization**: Sample probability densities initialized as $(1/N, \ldots, 1/N)^T$. Model parameters initialized as $\theta^{(0)}$.
**for** $i = 0$ **to** Epochs **do**
    1. $G_1 = g(G_0)$, detailed in 3.3.
    2. Simulate gradient flow for $T_{\text{in}}$ time steps according to Equation 3 and 4 to learn an approximate worst-case probability weight $q^{T_{\text{in}}}$.
    3. $\theta^{(i+1)} \leftarrow \theta^{(i)} - \gamma \nabla_\theta (\sum_i q_i^{T_{\text{in}}} \ell(f_\theta(x_i), y_i))$
**end for**

---

involves a minimax optimization, where we iteratively perform gradient ascents (on $q$) for the inner maximization and descents (on $\theta$) for the outer minimization. The pseudo-code of our algorithm is shown in Algorithm 1.

**Inner maximization problem.** For easy notion, we define

$$\mathcal{L}(\theta, q) := \sum_{i=1}^{N} q_i \ell(f_\theta(x_i), y_i) - \beta \cdot \sum_{i=1}^{N} q_i \log q_i.$$

Generally, the goal of the inner maximization problem in Equation 1 is to maximize $\mathcal{L}(\theta, q)$ and to minimize the topology penalty $\mathcal{GW}_{G_0}^2(\hat{\mathbb{P}}_s, q)$ w.r.t. sample densities $q$. Instead of directly computing the topology penalty, we solve the inner maximization via gradient ascents on $q$ in the geometric Wasserstein space $(\mathcal{P}_o(G_0), \mathcal{GW}_{G_0})$, where the topology penalty $\mathcal{GW}_{G_0}^2(\hat{\mathbb{P}}_s, q)$ is approximated by the length of the gradient flow trajectory in the metric space.

As stated in Definition 2.1, the continuous gradient flow is denoted by $q : [0, 1] \rightarrow \mathcal{P}_o(G_0)$, and $q(t)$ represents the sample density at time $t \in [0, 1]$. In order to derive empirical optimization approaches, we introduce the *time-discretized* gradient flow, denoted by $q^\tau : [0, T] \rightarrow \mathcal{P}_o(G_0)$, and the superscript $\tau$ is the value of *time step* (here we introduce this superscript because different time steps refer to different time-discretized gradient flow function). For the approximate optimization, we leverage this time-discretized gradient flow (with time step $\tau$) of $-\mathcal{L}(\theta, q)$ in the geometric Wasserstein space $(\mathcal{P}_o(G_0), \mathcal{GW}_{G_0})$ as:

$$q^\tau(t + \tau) \leftarrow \arg \max_{q \in \mathcal{P}_o(G_0)} \mathcal{L}(\theta, q) - \frac{1}{2\tau} \cdot \mathcal{GW}_{G_0}^2(q^\tau(t), q),$$
$$(2)$$

which aims to obtain the "local" maximum of $\mathcal{L}(\theta, q)$ around $q^\tau(t)$ at time $t$ and restricts the topology distance $\mathcal{GW}_{G_0}^2(q^\tau(t), q)$. We derive the analytical form of Equation 2 as $\tau \rightarrow 0$. For the ease of notion, the sample density of the $i$-th node at time $t$, originally denoted by $q_i^\tau(t)$, is

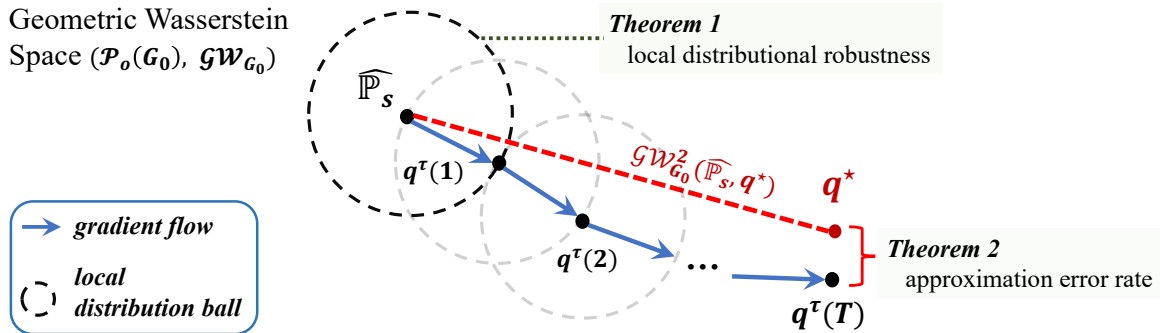

*Figure 2.* Illustration of the gradient flow in the geometric Wasserstein space $(\mathcal{P}_o(G_0), \mathcal{GW}_{G_0})$, where each point denotes a probability distribution in $\mathcal{P}_o(G_0)$, and the distance is measure by the discrete geometric Wasserstein distance. The black circle denotes the local distribution set around a distribution, and the blue arrow represents the one-step gradient flow. $q^\tau(T)$ denotes the approximated inner maximizer obtained by our algorithm, and $q^\star$ denotes the ground-truth inner maximizer (defined in Theorem 3.3). In Theorem 3.2, we demonstrate that the one-step gradient flow is equivalent to distributionally robust optimization around a local uncertainty set, and in Theorem 3.3, we characterize the approximation error rate between $q^\tau(T)$ and $q^\star$.

abbreviated as $q_i(t)$, and then Equation 2 becomes:

$$\frac{dq_i(t)}{dt} = \sum_{j:(i,j)\in E} w_{ij} v_{ij} \Big( \mathbb{I}(v_{ij} > 0) q_j + \mathbb{I}(v_{ij} \leq 0) q_i \Big)$$
$$v_{ij} = \ell_i - \ell_j + \beta(\log q_j - \log q_i), \quad \text{for } (i,j) \in E \tag{3}$$

where $E$ is the edge set of graph $G_0$, $w_{ij}$ is the edge weight between node $i$ and $j$, $\mathbb{I}(\cdot)$ is the indicator function, and $\ell_i$ represents the prediction error on the $i$-th node. Intuitively, $v_{ij}$ can be viewed as the transferring velocity of the sample density from node $j$ to node $i$.

Let $\lambda = \frac{1}{2\tau}$, Equation 2 exactly aligns with the goal of our inner maximization problem in Problem 1. Specifically, the original topology penalty calculates the distance $\mathcal{GW}_{G_0}^2(\hat{\mathbb{P}}_s, q^\tau(t))$ between $\hat{\mathbb{P}}_s$ and $q^\tau(t)$, and our gradient flow approximates it via $\sum_{i=1}^t \mathcal{GW}_{G_0}^2(q^\tau(i-1), q^\tau(i))$ (see blue curves in Figure 2). In Theorem 3.3, we characterize the error rate of this approximation.

*Remark* 3.1. Here we make some remarks on Equation 3:
(a) The gradient of the $i$-th node's probability density depends on its neighbors in graph $G_0$. This corresponds with our motivation that the reweighting scheme should incorporate topology information. Furthermore, since the transfer is between neighbors, the probability density $p$ remains **locally smooth** w.r.t. the graph structure (or manifold), which avoids overemphasis on some noisy samples.
(b) Combined with our topology penalty, the entropy penalty acts as a **non-linear graph Laplacian operator** to further the smoothness of probability densities along the manifold.
(c) The gradient flow in Equation 3 is implemented by message propagation, which **scales linearly with sample size** and enjoys parallelization by GPU. For a detailed complexity analysis, please refer to Appendix F.

(d) Due to the random sampling of labeled nodes during training for node classification tasks, it means that for certain nodes we cannot compute the loss, which disrupts the connectivity and hinders the calculation of Equation 3, we intuitively set the loss for these unlabeled nodes to the mean loss of the labeled nodes, and this approach has proven to be adequate. For other potential solutions, please refer to the Appendix E.

Based on Equation 3, we can solve the inner maximization problem via gradient ascent as:

$$q_i(0) \leftarrow 1/N, \tag{4}$$
$$q_i(t+1) \leftarrow q_i(t) + \tau \cdot dq_i(t)/dt, \quad \text{for } i \in [N]. \tag{5}$$

In addition, we demonstrate the equivalence between Equation 2 and distributional robustness in Theorem 3.2, justifying how our proposed TAR can provide robustness against distribution shifts. And in Theorem 3.3, we characterize the error rate of our approximation as $e^{-CT_{\text{in}}}$, which allows a relatively accurate approximation with finite $T_{\text{in}}$ steps.

**Outer minimization problem.** For the outer minimization problem, we perform gradient descent on model parameters $\theta$. According to the overall objective in Equation 1, the loss function is simply a weighted average:

$$\theta^{(t+1)} \leftarrow \theta^{(t)} - \gamma \cdot \nabla_\theta \Big( \sum_{i=1}^N q_i(T_{\text{in}}) \cdot \ell(f_\theta(x_i), y_i) \Big), \tag{6}$$

where $\gamma$ is the learning rate, and $q_i(T_{\text{in}})$ denotes the probability density of the $i$-th node (after $T_{\text{in}}$ steps gradient flow).

### 3.2. Theoretical Analysis

In this section, we investigate in-depth our proposed optimization algorithm. As illustrated in Figure 2, we first

*Table 1.* The performance on OOD benchmark datasets under Covariate Shift , using GCN as the backbone. We report the average test accuracy (except for Twitch, where we use ROC-AUC as the evaluation metric) and standard deviations over 5 runs. The best results are shown in **bold**, and the second best results are shown in underline. OOM denotes out of memory.

| Dataset | | CBAS | WebKB | Twitch | Cora | | Arxiv | | Require domain information |
|---|---|---|---|---|---|---|---|---|---|
| Domain | | color | university | language | word | degree | time | degree | |
| Base | ERM | 78.29±3.10 | 19.37±6.01 | 50.04±2.53 | 64.72±0.54 | 55.33±0.90 | 71.48±0.30 | 57.59±0.36 | No |
| Invariant Learning | IRM | 78.00±3.73 | 21.59±9.30 | 50.97±3.62 | 64.70±0.45 | 55.34±0.88 | 71.36±0.13 | 57.54±0.15 | Yes |
| | VREx | 78.57±2.02 | 36.83±1.99 | 51.26±4.82 | 65.02±0.45 | 55.44±0.78 | 71.58±0.28 | 57.60±0.27 | Yes |
| Domain Generalization | Coral | 78.00±3.59 | 30.16±5.67 | 53.46±0.41 | 64.85±0.33 | 55.21±0.90 | 71.48±0.40 | 57.23±0.16 | Yes |
| | DANN | 77.71±3.13 | 33.49±9.61 | 50.47±3.14 | 64.72±0.39 | 55.32±0.92 | 71.68±0.19 | 57.25±0.23 | No |
| Graph OOD | SRGNN | 76.86±1.56 | 22.86±10.79 | 49.71±1.81 | 64.63±0.34 | 55.22±1.05 | 70.78±0.18 | 57.53±0.24 | Yes |
| | EERM | 75.14±3.29 | 33.97±11.42 | OOM | 64.88±0.38 | 55.30±0.89 | OOM | OOM | No |
| | FLOOD | 83.14±4.21 | 33.97±3.09 | 55.14±1.78 | 64.91±0.45 | 54.78±0.82 | 71.75±0.37 | 58.90±0.22 | No |
| | CIT | 80.86±2.96 | 28.89±9.09 | OOM | 64.34±0.72 | 54.74±0.82 | OOM | OOM | No |
| Sample Reweighting | KL-DRO | 77.71±1.63 | 32.06±16.96 | 53.76±4.19 | 64.85±0.51 | 55.14±0.87 | 71.58±0.24 | 57.59±0.30 | No |
| | GroupDRO | 77.14±2.67 | 25.24±9.57 | 50.48±2.04 | 64.95±0.59 | 55.05±0.83 | 71.46±0.33 | 57.25±0.23 | Yes |
| **Ours** | **TAR** | **87.43±5.09** | **37.46±4.84** | **57.82±2.13** | **65.64±0.37** | **56.62±0.23** | **71.99±0.23** | **59.65±0.14** | **No** |

prove that each step of the gradient flow exactly finds the worst-case distribution within a local uncertainty set (see black circle in Figure 2).

**Theorem 3.2** (Distributional robustness). *For any* $\gamma > 0, t > 0$ *and given* $\theta$, *denote the solution of Equation 2 as*

$$q^\star = \arg \max_{q \in P_o(G_0)} \mathcal{L}(\theta, q) - \gamma \mathcal{GW}^2_{G_0}(p, q).$$

*Let* $\epsilon = \mathcal{GW}^2_{G_0}(p, q^\star)$, *we have*

$$\underbrace{\max_{q \in P_o(G_0)} \mathcal{L}(\theta, q) - \gamma \mathcal{GW}^2_{G_0}(p, q)}_{\text{one-step gradient flow at time } t} =$$

$$\underbrace{\max_{q: \mathcal{GW}^2_{G_0}(p,q) \leq \epsilon} \mathcal{L}(\theta, q)}_{\text{the worst-case distribution within a local distribution set}}. \quad (7)$$

*The proof can be found in Appendix G.*

Theorem 3.2 shows that, for the inner maximization, our proposed gradient flow is equivalent to finding the worst-case distribution within a small distribution set. Therefore, the weighted average loss function in Equation 6 captures the worst-case distribution that may occur in testing, which shares the similar idea with distributionally robust optimization (Duchi & Namkoong, 2018; Blanchet et al., 2019; Liu et al., 2022). This demonstrates the strength of our proposed TAR framework in dealing with potential distribution shifts.

Then based on the results in (Chow et al., 2017, Theorem 5) and (Liu et al., 2022, Theorem 3.2), we move on to analyze the error rate of our approximation in Theorem 3.3.

**Theorem 3.3** (Approximation error rate). *Given the GNN parameter* $\theta$, *denote the approximate sample densities in Equation 3 after* $T_{in}$ *steps of gradient flow as* $q(T_{in})$, *and* $\epsilon = \mathcal{GW}^2_{G_0}(\hat{\mathbb{P}}_s, q(T_{in}))$ *is the geometric Wasserstein distance*

*from the original source distribution. Denote the ground-truth worst-case distribution with the same distance* $\epsilon$ *as:*

$$q^\star = \arg \max_{q: \mathcal{GW}^2_{G_0}(\hat{\mathbb{P}}_s, q) \leq \epsilon} \mathcal{L}(\theta, q),$$

*Then we have:*

$$\frac{\mathcal{L}(\theta, q(T_{in})) - \mathcal{L}(\theta, \hat{\mathbb{P}}_s)}{\mathcal{L}(\theta, q^\star) - \mathcal{L}(\theta, \hat{\mathbb{P}}_s)} > 1 - e^{-CT_{in}}, \quad (8)$$

*where* $C > 0$ *is a constant and its value depends on the loss function* $\ell$, *hyper-parameter* $\beta$, *and sample size* $N$. *The proof can be found in Appendix G.*

*Remark* 3.4. We make some remarks here:
(1) Since the goal of our reweighting is to maximize $\mathcal{L}(\theta, q)$ w.r.t. $q$, we utilize the increase of $\mathcal{L}$ to characterize how "approximate" is our optimization. In Equation 8, the denominator of the left-hand side represents the maximal increase, and the numerator is the increase attained through our approximation. As the ratio approaches 1.0, our approximation becomes increasingly precise.
(2) Our theoretical results show that the error rate is $e^{-CT_{in}}$, which shrinks fast as the number of time step $T_{in}$ increases. This further demonstrates that our optimization is able to find good approximations in finite (usually small) number of gradient flow steps.

### 3.3. Graph Extrapolation

In the previous section, we proved that gradient flow is equivalent to distributionally robust optimization around a local uncertainty set. Given that the source distribution is fixed, the scope of the uncertainty set is inherently limited. To address this limitation, we propose a **Graph Extrapolation (GE)** approach $g(\cdot)$ to expand the source distribution, thereby enlarging the uncertainty set and further enhancing robustness against potential distribution shifts. Specifically,

we adopt two graph data augmentation techniques, DropEdge (Rong et al.) and MaskFeature (You et al., 2020), to expand the source data distribution i.e. generate a new graph $G_1$ according to $G_1 = g(G_0)$. During training, we first conduct GE to the input graph to construct an augmented graph at the beginning of each epoch, then conduct TAR on the augmented graph to train the model.

## 4. Experiment

We conduct experiments on five widely used node classification datasets from GOOD benchmark (Gui et al., 2022) to validate the effectiveness of TAR in improving out-of-distribution (OOD) generalization.

### 4.1. Datasets and Baselines

**Datasets.** We use five node classification datasets under both concept shift and covariate shift (the detailed definition of these two shift are provided in Appendix B) : WebKB (Pei et al., 2020), CBAS (Ying et al., 2019), Twitch (Rozember-czki & Sarkar, 2020), Cora (Bojchevski & Günnemann, 2017) and Arxiv (Hu et al., 2020). We followed the GOOD benchmark (Gui et al., 2022) for data splitting, a standard widely adopted in prior research (Sui et al., 2023; Liu et al., 2023; Guo et al., 2024). The statistics of these datasets and detailed information are provided in Appendix B.

**Baselines.** We compare our proposed TAR method with several commonly used baseline methods using GCN (Kipf & Welling, 2017) as the backbone (we also provide results using recently proposed sota graph transformer polynormer (Deng et al., 2024) as backbone, please refer to Appendix D). These include Empirical Risk Minimization (**ERM**) and two general invariant learning methods: **IRM** (Arjovsky et al., 2019) and **VREx** (Krueger et al., 2021). Additionally, we compare with two sample reweighting methods, **KL-DRO** (Hu & Hong, 2013) and **Group DRO** (Sagawa et al., 2019). Furthermore, we evaluate our method against two common domain generalization approaches: **DANN** (Ganin et al., 2016) and **Deep Coral** (Sun & Saenko, 2016), as well as four graph-specific domain generalization baselines: **EERM** (Wu et al., 2021), **SRGNN** (Zhu et al., 2021), **FLOOD** (Liu et al., 2023), and **CIT** (Xia et al., 2024). Among these methods, only KL-DRO, DANN, EERM, FLOOD, and CIT do not require domain labels to address distribution shifts. Similarly, our proposed TAR method does not require domain labels. Please refer to Appendix C for more implementation details.

### 4.2. Performance Comparison

Table 1 and 2 summarize the results of our method and other baselines on four datasets under both covariate shift and concept shift. We have the following observations:

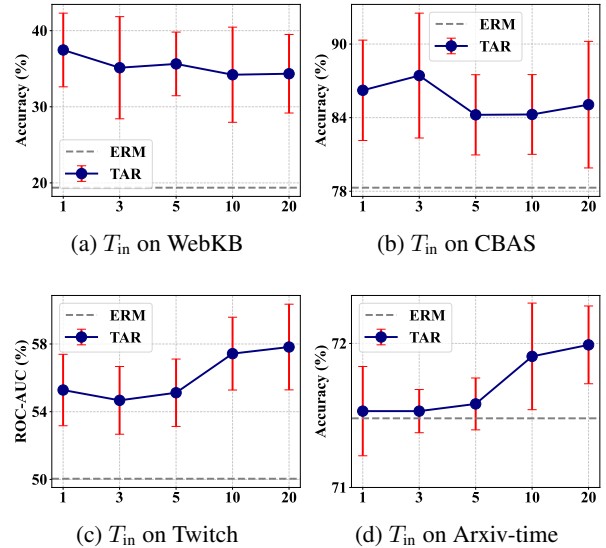

(a) $T_{in}$ on WebKB      (b) $T_{in}$ on CBAS

(c) $T_{in}$ on Twitch      (d) $T_{in}$ on Arxiv-time

*Figure 3.* The effects of $T_{in}$ (the number of gradient flow) of our proposed TAR algorithm, we provide the results under covariate shift.

Graph OOD methods generally achieve better results, as they obtain the second-best performance in 7 out of 14 settings. We attribute this to their ability to account for graph topology when addressing distribution shifts, leading to improved outcomes. This highlights the necessity of incorporating graph topology information when tackling node classification tasks. Additionally, VREx also performs well, achieving the second-best results in 5 out of 14 settings. However, this method relies on domain information (i.e., environment labels), which is often difficult to obtain in real-world applications, thereby limiting its applicability. Group DRO achieves the second-best results in the remaining two settings, demonstrating the robustness of sample reweighting methods. However, since it does not consider graph topology, its performance improvement remains limited.

As shown in Tables 1 and 2, our proposed method, TAR, consistently outperforms other baselines across all datasets under both concept shift and covariate shift. This highlights the effectiveness of TAR in addressing different types of distribution shifts. In particular, for the CBAS dataset, TAR achieves performance improvements of 4.29% and 3.00% under covariate shift and concept shift, respectively, compared to the best-performing baseline. Similarly, for the Twitch dataset, TAR demonstrates improvements of 2.68% and 1.45%, respectively, over the best-performing baseline. These results underscore the superiority of our method and emphasize the importance of incorporating topological information when addressing distribution shift challenges in node classification tasks.

*Table 2.* The performance on OOD benchmark datasets under Concept Shift, using GCN as the backbone. We report the average test accuracy (except for Twitch, where we use ROC-AUC as the evaluation metric) and standard deviations over 5 runs. The best results are shown in **bold**, and the second best results are shown in underline. OOM denotes out of memory.

| Dataset | | CBAS | WebKB | Twitch | Cora | | Arxiv | | Require domain information |
|---|---|---|---|---|---|---|---|---|---|
| Domain | | color | university | language | word | degree | time | degree | |
| Base | ERM | 82.43±2.56 | 27.16±0.93 | 51.59±3.63 | 64.03±0.28 | 60.30±0.46 | 65.64±0.27 | 54.81±0.40 | No |
| Invariant Learning | IRM | 82.00±1.88 | 26.06±0.73 | 49.78±3.27 | 63.93±0.35 | 60.26±0.49 | 65.54±0.34 | 56.72±0.30 | Yes |
| | VREx | 82.86±2.42 | 26.61±1.42 | 55.75±1.37 | 64.03±0.29 | 60.53±0.39 | 65.92±0.14 | 56.68±0.35 | Yes |
| Domain Generalization | Coral | 81.57±1.59 | 28.07±2.64 | 51.80±3.32 | 64.04±0.31 | 60.30±0.45 | 65.79±0.50 | 55.14±0.24 | Yes |
| | DANN | 83.57±1.52 | 29.36±3.31 | 51.67++3.50 | 63.96±0.29 | 60.23±0.51 | 65.67±0.42 | 55.34±0.45 | No |
| Graph OOD | SRGNN | 82.14±2.12 | 26.42±1.78 | 51.58±3.64 | 63.96±0.34 | 60.27+0.35 | 65.64±0.34 | 55.08±0.25 | Yes |
| | EERM | 65.71±0.90 | 29.91±0.50 | OOM | 63.42±0.35 | 60.21±0.55 | OOM | OOM | No |
| | FLOOD | 84.29±1.82 | 28.62±6.29 | 54.22±3.92 | 64.01±0.26 | 60.31±0.48 | 65.66±0.26 | 58.59±1.56 | No |
| | CIT | 83.71±0.60 | 28.99±2.11 | OOM | 63.77±0.36 | 60.05±0.59 | OOM | OOM | No |
| Sample Reweighting | KL-DRO | 81.14±1.48 | 29.54±1.37 | 51.87±3.16 | 64.03±0.32 | 60.52±0.22 | 65.51±0.16 | 54.70±0.35 | No |
| | GroupDRO | 82.71±1.78 | 29.17±1.76 | 52.24±4.05 | 64.10±0.38 | 60.43±0.40 | 65.93±0.24 | 56.24±0.44 | Yes |
| **Ours** | **TAR** | **87.29±0.78** | **30.83±1.90** | **57.20±3.97** | **64.73±0.23** | **61.73±0.22** | **66.08±0.23** | **59.26±1.40** | **No** |

*Table 3.* Results of ablation atudy

| Dataset | CBAS | | Twitch | |
|---|---|---|---|---|
| Shift | concept | covariate | concept | covariate |
| TAR | **87.29±0.78** | **87.43±5.09** | **57.20±3.97** | **57.82±2.13** |
| w/o SR | 83.57±1.52 | 84.29±4.52 | 49.78±3.27 | 55.14±1.78 |
| w/o GE | 83.14±3.14 | 79.43±1.63 | 55.39±3.35 | 54.54±4.70 |
| ERM | 82.43±2.56 | 78.29±3.10 | 51.59±3.63 | 50.04±2.53 |

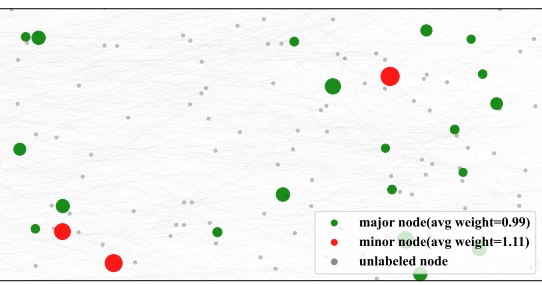

*Figure 4.* Learned node weight of TAR on CBAS under concept shift. The size of each node is proportional to its sample weight. TAR assigned greater node weight to these minor nodes.

### 4.3. Hyper-Parameter Analysis

We analyze the impact of the key parameters proposed in Algorithm 1: $T_{in}$ , its value determines the number of iterations over which the sample weights propagate along the graph edges. A larger $T_{in}$ allows weights to transfer over a broader range, whereas a smaller $T_{in}$ limits weight propagation to a few neighboring hops. As illustrated in Figure 3, performance reaches the best at $T_{in} = 1$ for the WebKB dataset and at $T_{in} = 3$ for the CBAS dataset. However, for the Twitch and Axiv datasets, performance continues to improve until $T_{in} = 20$. We attribute this to the larger size of these two datasets. Larger graphs may require more iterations to effectively propagate sample weights.

### 4.4. Ablation Study

We conducted an ablation study to investigate the effects of different modules in TAR, including the sample reweighting procedure (SR) and Graph Extrapolation module (GE). As shown in Table 3, we compared the TAR with its two variants "w/o SR" and "w/o GE" based on whether the SR and GE were enabled. Both modules contribute to the final result. Specifically, on the Twitch dataset, when SR is not used and only GE is applied, the performance is even lower than that of ERM. This highlights the importance of TAR's adjustment of sample weights in enhancing robustness.

### 4.5. Visualization

We conducted visualization experiments on the CBAS dataset under concept shift to provide an intuitive explanation for the effectiveness of our method. In this dataset, there is a spurious correlation between most nodes (major nodes) and their labels, while a small number of nodes (minor nodes) do not exhibit this spurious correlation. If the model primarily learns these spurious correlations from the major nodes during training, its performance will degrade when tested on a distribution that does not contain such correlations. As shown in Figure 4, TAR assigns higher training weights to these minority nodes that do not exhibit the spurious correlation, encouraging the model to focus more on these nodes and avoid learning the spurious correlation. As a result, TAR achieves a 4.86% improvement over ERM.

### 5. Conclusion

Through this work, we innovatively propose the Topology-Aware Dynamic Reweighting (TAR) framework to address

the distribution shift problem in node classification tasks. TAR utilizes a minimax approach to enhance the generalization ability of GNN models, incorporating topological structure information through gradient flows along graph edges. We further conduct theoretical analysis to reveal the ability of TAR to enhance the distributional robustness of GNNs. Experimental results confirm the effectiveness of TAR node classification under distribution shifts. Our TAR opens a new direction for addressing the distribution shift problem for node classification tasks.

## Impact Statement

This paper presents work whose goal is to advance the field of Graph Out of Distribution Generalization. There are many potential societal consequences of our work, none which we feel must be specifically highlighted here.

## Acknowledgements

This work is supported by grant 62471133 National Natural Science Foundation of China. This work is also supported by grant 2242024K40020 Central University Basic Research Fund of China, and the Big Data Computing Center of Southeast University. Peng Cui is supported by NSFC (No. 62425206, 62141607) and Beijing Municipal Science and Technology Project (No. Z241100004224009).

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

## A. Related Works

**Out of Distribution Generalization.** Out-of-Distribution (OOD) generalization aims to address the challenge of ensuring model robustness and generalization when faced with data that differ from the training distribution. Numerous studies have been dedicated to addressing the problem of OOD generalization, leading to the development of various methods for tackling OOD issues(Liu et al., 2021b). By accurately identifying the causal relationships between features and their corresponding labels, causal learning methods are expected to perform well even when the data distribution changes, as the underlying causal structure is often assumed to remain invariant across different environments or domains. Shifting the focus from strict causality to invariance, invariant learning aims to develop a representation or model that remains consistent across various environments. Invariant Risk Minimization (IRM)(Arjovsky et al., 2019) and Variance Risk Extrapolation (VREx) (Krueger et al., 2021) are two prominent methods specifically designed to address these challenges. IRM focuses on learning invariant features by ensuring that the optimal classifier remains the same across different environments, whereas VREx aims to minimize the variance of risks across environments, ensuring stable performance under distributional shifts. Another line of research focused on addressing OOD generalization problems involves distributionally robust optimization methods. These model-agnostic techniques come with strong theoretical guarantees and achieve OOD generalization by incorporating distributional robustness into the training process. This ensures that the model's performance remains stable across different data distributions. KL-DRO (Duchi & Namkoong, 2018) minimizes the KL divergence between training and potential test distributions. WDRO (Chen & Paschalidis, 2018; Sinha et al., 2017) leverages the Wasserstein distance to ensure robustness to distributional changes. Group DRO (Sagawa et al., 2019) aims to provide consistent performance across different subgroups by minimizing the worst-case risk among them. While invariant learning has been extensively applied in graph tasks(Wu et al., 2021; Xia et al., 2024; Li et al., 2022; Sui et al., 2023; Wu et al., 2022), there is relatively less application of distributionally robust optimization methods in graph tasks. Applying these methods to graphs requires addressing the unique structural properties of graphs, posing challenges that are specific to graph data.

**Graph Invariant Learning.** Recently, graph invariant learning has shown enormous success in addressing graph out-of-distribution problems(Wu et al., 2022; 2021; Sui et al., 2023; Zhou et al., 2022; Li et al., 2022; Xia et al., 2024). Graph invariant learning aims to exploit the invariant relationships between graph features(which can be divided into topological structures and node features) and labels across distribution shifts, while filtering out the variant spurious correlations caused by the environment. Recently, many methods have been proposed for graph-level tasks. GIL (Li et al., 2022) captures the invariant relationships between predictive graph structural information and labels in a mixture of latent environments. DIR (Wu et al., 2022) selects a subset of causal rationales and conducts data augmentation to create multiple distributions to improve generalization. MoleOOD (Yang et al., 2022) enhances the robustness of molecule learning and infers the environment in a fully data-driven manner. AIA (Sui et al., 2023) generates new environments while preserving the original stable features during the augmentation process with adversarial strategies. Compared to research on graph-level ood, less attention has been paid to learning node-level representations under distribution shifts from the invariant learning perspective (Wu et al., 2021; Liu et al., 2023; Xia et al., 2024). EERM (Wu et al., 2021) leverages multiple context explorers that are adversarially trained to maximize the variance of risks from multiple virtual environments to learn a node invariant predictor. Instead, FLOOD (Liu et al., 2023) applies data augmentation to construct multiple environments and maximize the variance of risks from multiple virtual environments to learn a node invariant predictor. CIT (Xia et al., 2024) generates nodes across different clusters, significantly enhances the diversity of the nodes, and helps GNNs learn the invariant representations. However, this line of invariant learning typically focuses on specific types of invariance (e.g., subgraph invariance in graphs), which may not cover all possible shifts. Besides, due to the lack of environmental information in the real world, generating new samples might introduce bias or noise.

## B. Datasets

We define the distribution shift problem in the node classification task following (Gui et al., 2022) below. The joint data distribution can be decomposed as $\mathbb{P}(Y, X) = \mathbb{P}(Y|X)\mathbb{P}(X)$. The main causes of distribution shifts can be separated into two types of shifts:

- **Covariate shift** ($\mathbb{P}_s(Y|X) = \mathbb{P}_t(Y|X), \mathbb{P}_s(X) \neq \mathbb{P}_t(X)$): This indicates that the feature distribution differs between the source distribution and the target distribution.

- **Concept shift** ($\mathbb{P}_s(Y|X) \neq \mathbb{P}_t(Y|X), \mathbb{P}_s(X) = \mathbb{P}_t(X)$): This indicates that there are spurious statistical correlations in the source distribution that may not hold in the target distribution.

*Table 4.* Statistics of GOOD datasets

| Dataset | #Node | #Edge | #Class | #Feat | Domain |
|---------|-------|-------|--------|-------|--------|
| CBAS | 700 | 3962 | 4 | 4 | Color |
| WebKB | 617 | 1138 | 5 | 1703 | University |
| Twitch | 34120 | 892346 | 2 | 128 | Language |
| Cora | 19793 | 126842 | 70 | 8710 | Word/Degree |
| Arxiv | 169343 | 1166243 | 40 | 128 | Time/Degree |

In this paper, we use five OOD node classification datasets from GOOD benchmark (Gui et al., 2022), including WebKB, CBAS, Twitch, Cora, and Arxiv. The statistics of these datasets are shown in Table 4.

- **CBAS** is a synthetic dataset where the input graph is constructed by attaching 80 house-like motifs to a 300-node Barabási–Albert base graph. The task involves predicting the role of each node, including whether a node is the top, middle, or bottom of a house-like motif, or belongs to the base graph, resulting in a 4-class classification task. In CBAS, different node colors are used as features, so OOD algorithms must address both covariate shifts caused by node color differences and concept shifts driven by color-label correlations.

- **WebKB** is a university webpage network dataset. Each node corresponds to a webpage, with the words on the page serving as node features, and edges representing hyperlinks between webpages. The task is a 5-class classification problem, where the goal is to predict the category of each webpage. The dataset is split based on the domain of the university, meaning the classification is based on the content of the webpages and their link structures, rather than on university-specific attributes.

- **Twitch** is a gamer network dataset where nodes represent gamers, with their games serving as node features, and edges representing friendships between gamers. The task is a binary classification problem aimed at predicting whether a user streams mature content. The dataset is split based on user language, which implies that the prediction should not be influenced by the language the user uses.

- **Cora** is a citation network dataset derived from the full Cora dataset. It consists of a small-scale citation graph where nodes represent scientific publications and edges represent citation links. The task is a 70-class classification of publication types. The dataset is split based on two domain criteria: word and degree. The first domain, word diversity, is defined by the number of selected words in a publication, which is completely unrelated to the label. The second domain is node degree, implying that the popularity of a paper (as indicated by its citation count) should not influence its classification.

- **Arxiv** is a citation network adapted from OGB (Hu et al., 2020). The input is a directed graph representing the citation network among arXiv papers in the field of computer science (CS). Nodes correspond to arXiv papers, and directed edges represent citation links between them. The task is to predict the subject area of these CS papers, making it a 40-class classification problem. The dataset is split based on two domain criteria: time (publication year) and node degree.

## C. Implementation Details

In our experiments, we utilized GCN and a recently proposed sota graph transformer Polynormer (Deng et al., 2024) as the backbone models.

- For **GCN**, we configured the models with 3 layers, a hidden dimension of 300, a dropout rate of 0.5, and a learning rate of 0.01.

- For **Polynormer**, the layer of the local module is set to 5 and the global module is set to 1, with a hidden dimension of 512 and a learning rate of 0.001.

Throughout all experiments, we employed the Adam optimizer with a weight decay of 0. For all baselines, we conducted a grid search as defined by the GOOD Benchmark or their original papers and reported their best results. Note that the graph

OOD algorithm EERM (Wu et al., 2021) and CIT (Xia et al., 2024) encounters CUDA out of memory on Arxiv and Twitch datasets due to its high memory requirement.

The searching spaces for all the hyper-parameters of TAR are as follows.

- Entropy term $\beta$: {1, 0.1, 0.01, 0.001}.

- TAR inner learning rate $\gamma$: {0.1, 0.01, 0.001}.

- Gradient flow iterations $T_{in}$: {1, 3, 5, 10, 20}.

- Graph extrapolation ratio: {0.0, 0.2, 0.4}

**Software and Hardware.** Our implementation is under the architecture of PyTorch (Paszke et al., 2019) and PyG (Fey & Lenssen, 2019). All of our experiments are run on one GeForce RTX 3090 with 24GB. The detailed versions of some key packages are listed below:

- python: 3.8

- pytorch: 1.13.1

- pyg: 2.3.1

## D. Comparion on SOTA Graph Transformer

Apart from the result using GCNs as the backbone in Table 1 and 2, we also conducted comparative experiments using Polynormer (Deng et al., 2024) as the backbone on the WebKB, CBAS and Twitch datasets.

As shown in the Tabel 5, we have the following observations: Compared to Tables 1 and 2, Polynormer with ERM outperforms GCN in 4 out of 6 settings, demonstrating that Polynormer, as a recently proposed state-of-the-art Graph Transformer, exhibits superior generalization capabilities compared to GCN. Our proposed method, TAR, achieves better performance than all other baselines across all settings, particularly under concept shift on the Twitch dataset, where it outperforms the second-best method by a significant margin of 7.4%. Experiments conducted with both GCN and Polynormer as backbones further validate the **versatility** of our approach and its **superiority** over competing methods.

*Table 5.* The performance on three OOD benchmark datasets, using Polynormer (Deng et al., 2024) as the backbone. We report the average test accuracy (except for Twitch, where we use ROC-AUC as the evaluation metric) and standard deviations over 5 runs. The best results are shown in **bold**, and the second best results are shown in underline. OOM denotes out of memory.

| Dataset | | CBAS | | WebKB | | Twitch | | Require domain |
|---|---|---|---|---|---|---|---|---|
| Shift | | concept | covariate | concept | covariate | concept | covariate | information |
| Base | ERM | 88.14+2.35 | 98.29±1.20 | 42.57±3.59 | 14.60±4.48 | 51.02±1.23 | 54.06±1.41 | No |
| Invariant Learning | IRM | 87.86±3.23 | 97.14±2.12 | 42.57±5.25 | 13.81±1.44 | 51.07±1.15 | 54.08±1.33 | Yes |
| | VREx | 89.43±1.37 | 99.14±1.28 | 44.04±2.90 | 15.71±2.71 | 52.80±0.85 | 54.30±1.10 | Yes |
| Domain Generalization | Coral | 87.71±1.85 | 97.71±1.63 | 42.75±5.37 | 13.49±0.56 | 51.33±1.19 | 53.45±1.29 | Yes |
| | DANN | 87.86±2.7 | 96.57±2.96 | 44.95±1.30 | 14.29±3.81 | 50.91±1.33 | 54.27±1.37 | No |
| Graph OOD | SRGNN | 87.43±3.06 | 98.00±2.96 | 41.10±1.29 | 14.60±2.95 | 51.00±1.24 | 54.23±1.30 | Yes |
| | EERM | 85.71±2.81 | 97.14±2.67 | 43.49±3.92 | 25.71±7.89 | OOM | OOM | No |
| | FLOOD | 88.57±2.08 | 98.57±1.01 | 41.28±1.83 | 14.92±4.14 | 51.75±3.02 | 54.50±0.35 | No |
| | CIT | 89.57±0.81 | 98.29±1.20 | 43.12±4.54 | 14.76±3.30 | OOM | OOM | No |
| Sample Reweighting | KL-DRO | 88.36±1.30 | 98.86±1.56 | 43.49±2.56 | 29.37±14.64 | 55.86±5.00 | 54.46±1.42 | No |
| | GroupDRO | 88.71±1.37 | 99.14±1.92 | 43.49±3.35 | 14.29±1.59 | 53.62±3.52 | 54.21±1.26 | Yes |
| **Ours** | **TAR** | **91.14±1.20** | **99.43±1.78** | **47.89±1.76** | **30.48±7.81** | **63.25±1.84** | **56.69±0.88** | **No** |

## E. Comparative Analysis of Solutions for Disconnected Trained Nodes

During the training of node classification tasks, the labeled nodes are randomly sampled, which often results in these labeled nodes not being directly connected in the space. Consequently, the gradient defined in Equation 3 cannot be computed. In this paper, we sets the loss of the unlabeled nodes to the mean loss of the labeled nodes, we also try other solutions, as detailed below:

- **TAR.** This method sets the loss of the unlabeled nodes to the mean loss of the labeled nodes. This approach preserves the original graph structure, allowing the gradient flow between training nodes to occur indirectly through the unlabeled nodes.

- **TAR-N.** This method performs a breadth-first search for each labeled node during training to find $K$ reachable labeled nodes. Then, it adds edges between the current node and these $K$ nodes, creating a new graph, and performs the gradient flow on this new graph. This approach enables the gradient to propagate directly among the training nodes but disrupts the original graph structure.

We conduct experiments both on CBAS and Cora (degree) to compare their performance. The results are summarized in Table 6. TAR-N performed worse than TAR under all settings. This suggests that preserving the original graph structure might be more reliable. Additionally, TAR-N requires extra computational overhead for the breadth-first search compared to TAR.

*Table 6.* The performance of TAR with orginal graph or TAR with reconnected graph on graph OOD datasets.

| Dataset | CBAS | | Cora | |
|---------|------|------|------|------|
| Shift | concept | covariate | concept | covariate |
| ERM | 82.43±2.56 | 78.29±3.10 | 60.30±0.46 | 55.33±0.90 |
| TAR | **87.29±0.78** | **87.43±5.09** | **61.73±0.22** | **56.62±0.23** |
| TAR-N | 86.71±2.29 | 86.29±2.96 | 61.57±0.22 | 56.17±0.44 |

## F. Complexity Analysis

Our approach is model-agnostic, making it applicable to any graph neural network. Moreover, it does not involve any learnable parameters, ensuring high efficiency.

Below, we analyze its time complexity: For an $l$-layer, $d$-dimensional GCN on a graph with $n$ nodes and $m$ edges, the time cost of TAR includes: **Outer Training**: $O(ld^2n + lm)$, which encompasses feature transformation and neighbor aggregation. **Inner Loop**: The $k$ iterations have a time complexity of $O(kn + km)$, introducing minimal additional computation since $k \ll ld^2$. The inner loop of TAR operates in a message-passing style and is parameter-free, ensuring scalability comparable to GCNs for large-scale graphs.

We compared the time complexity of our method with other model-agnostic approaches, as shown in Table 7. It can be observed that our method, similar to Group DRO and VREx, introduces only **little overhead**, demonstrating excellent scalability. In comparison to CIT, our approach achieves faster training speeds and avoids out-of-memory (OOM) issues, making it more efficient and practical for large-scale applications.

*Table 7.* Time cost of different OOD methods (ms/epoch).

| Methods | ERM | GroupDRO | VREx | CIT | TAR (ours) |
|---------|-----|----------|------|-----|------------|
| CBAS | 10.1±0.2 | 10.3±1.2 | 10.8±0.2 | 15.0±4.3 | 12.0±2.2 |
| Arxiv | 181.8±1.0 | 232.9±1.4 | 231.8±0.8 | OOM | 235.1±3.1 |

# G. Proof

## G.1. Proof of Theorem 1

*Proof.* Denote $q^\star = \arg \sup\limits_{q \in P_o(G_0)} \mathcal{L}(\theta, q) - \frac{1}{2\tau}\mathcal{GW}^2_{G_0}(p, q)$, and we have $\epsilon = \mathcal{GW}^2_{G_0}(p, q^\star)$.

Then we prove by contradiction: we first assume

$$q' = \arg \sup\limits_{q:\mathcal{GW}^2_{G_0}(p,q) \leq \epsilon} \mathcal{L}(\theta, q)$$

, which indicates that $\mathcal{L}(\theta, q') \geq \mathcal{L}(\theta, q^\star)$ and $\mathcal{GW}^2_{G_0}(p, q') \leq \epsilon$. Therefore, we have $\mathcal{GW}^2_{G_0}(p, q') \leq \mathcal{GW}^2_{G_0}(p, q^\star)$. Denote

$$\mathcal{R}(\theta, q) = \mathcal{L}(\theta, q) - \frac{1}{2\tau}\mathcal{GW}^2_{G_0}(p, q)$$

, then we have $\mathcal{R}(\theta, q^\star) \leq \mathcal{R}(\theta, q')$. This leads to contradiction since $q^\star$ is the supremum point of $\mathcal{R}(\theta, \cdot)$. $\square$

## G.2. Proof of Theorem 2

*Proof.* Based on the (Chow et al., 2017, Theorem 5), we have

$$\mathcal{L}(q(\infty)) - \mathcal{L}(q(t)) \leq e^{-Ct}(\mathcal{L}(q(\infty)) - \mathcal{L}(q(0))),$$

where $C > 0$ is a constant depending on the loss function $\ell$, hyper-parameter $\beta$, and the sample size $N$.

Then we denote the real worst-case distribution within the $\epsilon$-radius discrete Geometric Wasserstein-ball as $q^*$, and we have

$$\mathcal{L}(q(\infty)) - \mathcal{L}(q^\star) + \mathcal{L}(p^\star) - \mathcal{L}(q(t)) \leq e^{-Ct}(\mathcal{L}(q(\infty)) - \mathcal{L}(q^\star) + \mathcal{L}(q^\star) - \mathcal{L}(q(0))).$$

Therefore, we have

$$\mathcal{L}(q^\star) - \mathcal{L}(q(t)) \leq e^{-Ct}(\mathcal{L}(q^\star) - \mathcal{L}(q(0))) - (1 - e^{-Ct})(\mathcal{L}(q(\infty)) - \mathcal{L}(q^\star)),$$

and

$$\frac{\mathcal{L}(q^\star) - \mathcal{L}(q(t))}{\mathcal{L}(q^\star) - \mathcal{L}(q(0))} \leq e^{-Ct} - (1 - e^{-Ct})\frac{\mathcal{L}(q(\infty)) - \mathcal{L}(q^\star)}{\mathcal{L}(q^\star) - \mathcal{L}(q(0))} < e^{-Ct}.$$

$\square$

# H. Limitations and Future Work

In this paper, we propose a model-agnostic node classification OOD generalization algorithm TAR that leverages the existing topological information in node classification tasks to achieve better local robustness. However, as discussed in Appendix E, due to the random sampling of labeled nodes in node classification tasks, the nodes with computed losses are not directly connected, with other nodes without computed losses in between. Although we explored two potential solutions in Appendix E, we consider these solutions suboptimal. In the future, we aim to design a more effective method to address this issue.

