# OpenReview forum: "Topology-Aware Dynamic Reweighting for Distribution Shifts on Graph"
_ICML.cc/2025/Conference — ICML 2025 poster_

### Official Review · Reviewer_RR32 · 2025-03-12

**Overall Recommendation:** 3

**Summary:**

This paper introduces Topology-Aware Dynamic Reweighting (TAR), a novel framework designed to enhance node classification performance under distribution shifts by leveraging graph topology. Unlike invariant learning approaches that rely on strict assumptions about environment labels, TAR applies a dynamic sample reweighting strategy that incorporates gradient flow along graph edges to adjust sample importance during training. The framework employs a minimax optimization process: the inner maximization learns probability densities for samples, while the outer minimization trains the model on a dynamically weighted distribution. Theoretical analysis demonstrates that TAR identifies the local worst-case distribution, improving robustness against unseen distributions. Experimental evaluations on multiple OOD node classification benchmarks show that TAR consistently outperforms existing methods, particularly in settings where domain information is unavailable.

**Claims And Evidence:**

TAR’s theoretical foundation ensures robustness and convergence：While the theory is sound, additional empirical ablations (e.g., impact of different loss functions) could strengthen the claim.

**Essential References Not Discussed:**

No

**Experimental Designs Or Analyses:**

The experiments are well-designed.

**Methods And Evaluation Criteria:**

The evaluation methodology is generally appropriate for the problem.

**Other Comments Or Suggestions:**

1. How does TAR compare to simpler DRO methods in terms of training time?
2. How does TAR perform on heterophilic graphs, where homophily assumptions do not hold?

**Other Strengths And Weaknesses:**

Strengths：
1. Unlike standard sample reweighting methods that treat training instances independently, TAR explicitly considers graph topology during reweighting. The use of gradient flow along graph edges ensures that sample weights are influenced by the structural context of nodes, making the approach more aligned with the underlying data distribution. This is particularly important in graphs where neighboring nodes share similar properties.

2. The inner maximization problem dynamically adjusts sample weights by optimizing against a local worst-case distribution, while the outer minimization problem trains the model using these reweighted samples. This approach is inspired by distributionally robust optimization (DRO) but extends it to a graph-based setting with topology-aware constraints. The theoretical justification for this method strengthens its credibility.

3. The paper provides rigorous theoretical analysis that supports the proposed method. These results ensure that the dynamic reweighting mechanism is theoretically grounded, rather than being an empirical heuristic.

Weaknesses:
1. While TAR improves robustness under distribution shifts, it is unclear how much performance degradation occurs on in-distribution (ID) data. Many robust optimization methods sacrifice ID accuracy to gain robustness, but the paper does not explicitly analyze this trade-off. A controlled study where ID and OOD accuracy are jointly evaluated would provide deeper insights into this aspect.

2. TAR’s effectiveness heavily relies on gradient flow over graph edges. However, in sparse graphs with low connectivity, the amount of information that can be propagated during sample reweighting may be severely limited. Conversely, in densely connected graphs, the method might smooth weights too aggressively. The paper does not systematically analyze how TAR performs across different graph sparsity levels or degree distributions, which could affect its generalizability.

3. The method optimizes a weighted loss function using cross-entropy loss, but it is unclear how TAR would perform under alternative loss functions (e.g., margin-based losses, contrastive losses). Some OOD methods are highly sensitive to loss function choice, and the paper does not explore whether TAR is robust to such variations.

**Questions For Authors:**

Regarding the questions, please refer to the weaknesses section in the strengths and weaknesses part.

**Relation To Broader Scientific Literature:**

The paper is well-positioned within the OOD generalization and graph representation learning literature.

**Theoretical Claims:**

Yes

---

> ### Author Rebuttal · Authors · 2025-03-31
>
> We sincerely appreciate your careful review and insightful feedback. Below we provide point-by-point responses to address all raised concerns.
>
> # Performance when there is no shift during testing
> Thank you for your insightful suggestions!
>
> We compare TAR with several typical robust optimization methods, and the results are presented in the table below. Our key observations are as follows:
>
> - On the CBAS and Twitch datasets, TAR consistently outperforms baselines in both IID (no shift) and OOD (covariate shift) settings. Notably, the performance improvements under OOD are significantly more pronounced than those under IID.
> - Some methods, such as VREx, while achieving notable gains on CBAS and Twitch under the OOD setting, exhibit substantial performance degradation in the IID setting.
> - On the Cora dataset, TAR shows a slight, statistically insignificant drop in performance under IID. However, it achieves the best performance under the OOD setting, surpassing all other methods.
>
> These results highlight TAR's robustness and effectiveness across different scenarios.
>
> |dataset|CBAS|CBAS|Twitch|Twitch|Cora|Cora|
> |-|-|-|-|-|-|-|
> |shift|IID|OOD|IID|OOD|IID|OOD|
> |ERM|86.00±1.20|78.29±3.10|72.83±1.23|50.04±2.53|69.65±0.55|64.72±0.54|
> |VREx|83.14±3.26|78.57±2.02|70.84±4.15|51.26±4.82|69.63±0.31|65.02±0.45|
> |KLDRO|86.00±1.86|77.71±1.63|73.35±0.63|53.76±4.19|69.64±0.62|64.85±0.51|
> |GroupDRO|86.29±1.28|77.14±2.67|71.54±3.34|50.48±2.04|**69.77±0.26**|64.95±0.59|
> |TAR|**90.29±2.93**|**87.43±5.09**|**73.73±0.44**|**57.82±2.13**|69.53±0.77|**65.64±0.37**|
>
> # Performance of different graph sparsity levels
> Thank you for your question! As shown in the table below, the five benchmark datasets we employed exhibit diverse degree distributions, with average node degrees ranging from 1.52 to 26.15. Our method demonstrates consistently strong performance across all these varying degree distributions as you can see in Tables 1 and 2 in our paper.
> |dataset|webkb|cbas|twitch|cora|arxiv|
> |-|-|-|-|-|-|
> |avg. degree|1.52|5.66|26.15|6.41|13.67|
>
> # Performance on heterophilic graphs
> |dataset|webkb|cbas|twitch|cora|arxiv|
> |-|-|-|-|-|-|
> |avg. node homophilic|**0.14**|0.60|0.64|0.59|0.64|
>
> Thank you for your insightful question! As shown in the table above, our evaluation covers datasets with varying levels of homophily. Notably, WebKB represents a standard heterophilic graph benchmark.
> We additionally contruct an another OOD dataset chameleon, following the same priciple of GOOD benchmark [1] and using node degree as split type. As demonstrated in the results table below, TAR consistently achieves top performance on both heterophilic graphs, further validating its robustness across different homophily regimes.
>
> |dataset|WebKB|WebKB|Chameleon|Chameleon|
> |-|-|-|-|-|
> |shift|concept|covariate|concept|covariate|
> |ERM|27.16±0.93|19.37±6.01|34.65±2.56|33.13±4.53|
> |VRex| 26.61±1.42|36.83±1.99|34.81±2.51|**36.88±3.43**|
> |CIT|28.99±2.11|28.89±9.09|36.76±3.43|35.11±2.35|
> |GroupDRO|29.17±1.76|25.24±9.57|34.97±0.95|36.15±5.19|
> |TAR|**30.83±1.90**|**37.46±4.84**|**38.34±1.48**|**36.88±2.66**|
>
> # Performance on contrastive loss
> Thank you for your insightful question! TAR can be applied to different loss type. We apply TAR to two graph contrastive learning methods——GRACE [2] and COSTA [3]——to verify its effectiveness on contrastive loss. Specifically, we apply TAR in the pretrain stage and keep the finetune stage the same as the raw contrastive learning method. As shown in the table below, TAR consistenly improves the performance of different graph contrastive learning under distribution shifts.
> |dataset|WebKB|WebKB|CBAS|CBAS|
> |-|-|-|-|-|
> |shift|concept|covariate|concept|covariate|
> |ERM|27.16±0.93|19.37±6.01|82.43±2.56|78.29±3.10|
> |GRACE|32.66±5.79|19.68±9.73|90.29±1.64|93.71±1.28|
> |+ TAR|36.33±5.79|26.51±3.00|91.00±0.64|94.57±2.56|
> |COSTA|32.11±4.00|16.03±5.09|82.71±1.63|76.29±3.99|
> |+ TAR|33.39±3.28|19.68±5.68|85.29±2.45|79.14±3.73||
>
> [1] S. Gui, et al., Good: Agraph out-of-distribution benchmark. Thirty-sixth Conference on Neural Information Processing Systems Datasets and Benchmarks Track.
>
> [2] Y. Zhu, et al. Deep Graph Contrastive Representation Learning. ICML Workshop on Graph Representation Learning and Beyond, 2020.
>
> [3] Y. Zhang, et al. Costa: Covariance-preserving feature augmentation for graph contrastive learning. SIGKDD 2022.
>
> # Training Cost Analysis
> We compared the time cost (epoch/s) of our method with other methods, as shown in the table below. It can be observed that our method, similar to Group DRO and KLDRO, introduces only little overhead, demonstrating **excellent scalability**. For a comprehensive complexity analysis, please see Appendix F of our paper.
>
> |Methods|ERM|KLDRO|GroupDRO|CIT|TAR(ours)|
> |-|-|-|-|-|-|
> |CBAS (4k edges)|10.1±0.2|10.2±0.7|10.3±1.2|15.0±4.3|12.0±2.2|
> |Arxiv (1.2M edges)|181.8±1.0|186.9±1.1|232.9±1.4|OOM|235.1±3.1|
>
> Should you have any additional concerns, please do not hesitate to let us know.

---

> > ### Comment · Reviewer_RR32 · 2025-04-07
> >
> > I confirm that I have read the author's response to the question I raised, and I still keep my score.

---

### Official Review · Reviewer_sMqu · 2025-03-12

**Overall Recommendation:** 3

**Summary:**

This paper proposes a Topology-Aware Dynamic Reweighting (TAR) framework to address distribution shifts in node classification tasks using Graph Neural Networks (GNNs). Addressing the limitations of existing invariant learning methods (which rely on strong invariance assumptions) and sample reweighting approaches (which ignore graph topology), TAR dynamically adjusts node weights through gradient flows along graph edges while incorporating topological constraints via discrete geometric Wasserstein distance. The framework employs a minimax optimization strategy: the inner maximization identifies local worst-case distributions through constrained gradient flows, while the outer minimization trains the GNN model on these adversarially weighted samples to enhance distributional robustness. Theoretical analysis proves that TAR achieves exponential convergence rates and provides formal distributional robustness guarantees.

**Claims And Evidence:**

The claims in the submission are ​largely supported by clear and convincing evidence.

**Essential References Not Discussed:**

No

**Experimental Designs Or Analyses:**

Yes

**Methods And Evaluation Criteria:**

Yes

**Other Comments Or Suggestions:**

1. Please add the analysis of graph extrapolation.
2. It is suggested that the authors supplement the performance of the method in this paper when there is no covariance shift or concept shift during testing.
3. It is suggested that the authors provide sensitivity analyses for the following parameters: the coefficient of the entropy term, the TAR inner learning rate, the graph extrapolation ratio, and the coefficient of the topology penalty.

**Other Strengths And Weaknesses:**

Strengths：
1. Combines ​discrete geometric Wasserstein constraints with entropy regularization, ensuring smoothness.
2. Provides formal guarantees for ​distributional robustness and ​exponential convergence rates.
3. Demonstrates ​consistent improvements over most baselines across diverse datasets.
4. Maintains computational efficiency via message-passing mechanisms, with minimal overhead compared to ERM training.

Weaknesses:
1. Limited Analysis of Graph Extrapolation (GE): (1) Compare GE to other augmentation strategies (2) Analyze how GE interacts with TAR’s reweighting mechanism.
2. While TAR is efficient for moderate-sized graphs, the ​gradient flow iterations may scale poorly for graphs with billions of edges.
3. The bi-level optimization of min and max may lead to high training costs and instability.

**Questions For Authors:**

1. How is the sensitivity of other parameters？
2. Why does this paper additionally consider concept shift?

**Relation To Broader Scientific Literature:**

It offers a new perspective for graph node classification under distribution shifts.

**Theoretical Claims:**

The core theoretical claims are ​mathematically consistent within the stated assumptions.

---

> ### Author Rebuttal · Authors · 2025-03-31
>
> We sincerely appreciate your careful review and insightful feedback. Below we provide point-by-point responses to address all raised concerns.
>
> # Analysis of Graph Extrapolation
> ## 1.Interaction between GE and Reweighting
> Graph Extrapolation (GE) is a crucial component of TAR. Without GE, TAR reweighting can only simulate potential worst-case distributions by adjusting sample weights of existing data, which is limited by the coverage of current graph. In contrast, GE expands the potential distribution by transforming the graph. Specifically, while OOD samples may not appear during training, GE may construct such samples, allowing TAR reweighting to assign them higher weights and simulate potential distribution shifts.
>
> ## 2.Comparison with Mixup
> We use both Drop Edge and Mask Feature as GE since they transform the graph's topology and node features, respectively. These techniques are also utilized in other graph OOD methods like FLOOD. For comparison, we replace GE with Mixup, which only interpolates node features. The results show that GE consistently outperforms Mixup.
>
> |dataset|CBAS|Twitch|WebKB|
> |-|-|-|-|
> |Mixup|74.86±1.63|52.23±1.21|31.59±10.53|
> |TAR+mixup|78.00±2.78|54.51±1.30|36.51±8.57|
> |TAR+GE (ours)|**87.43±5.09**|**57.82±2.13**|**37.46±4.84**|
>
> # Performance when there is no shift during testing
> Thank you for your suggestion! Below are the comparative results and key observations:
> - On CBAS and Twitch datasets, TAR consistently outperforms baselines in both IID (no shift) and OOD (covariate shift) settings, with significantly larger gains under covariate shift.
> - On Cora dataset, TAR shows a slight, statistically insignificant drop in IID performance but achieves the best results under OOD, surpassing other methods.
>
> |dataset|CBAS|CBAS|Twitch|Twitch|Cora|Cora|
> |-|-|-|-|-|-|-|
> |shift|IID|OOD|IID|OOD|IID|OOD|
> |ERM|86.00±1.20|78.29±3.10|72.83±1.23|50.04±2.53|69.65±0.55|64.72±0.54|
> |VREx|83.14±3.26|78.57±2.02|70.84±4.15|51.26±4.82|69.63±0.31|65.02±0.45|
> |GroupDRO|86.29±1.28|77.14±2.67|71.54±3.34|50.48±2.04|**69.77±0.26**|64.95±0.59|
> |TAR|**90.29±2.93**|**87.43±5.09**|**73.73±0.44**|**57.82±2.13**|69.53±0.77|**65.64±0.37**|
>
> # Sensitivity analysis of other parameters
> Thank you for your insightful suggestion!
> We want to clrify that "the coefficient of the topology penalty" is not a hyperparameter, we set it to $\frac{1}{2\tau}$ in our optimization (page 3, line 163), and $\tau$ is exactly the TAR inner learning rate (page 4, equation 5). Due to space constraints, additional analysis figures for other parameters are provided in this anonymous [link](https://anonymous.4open.science/r/ICML-rebuttal-16C3). Please refer to it for more details.
>
> # Training Cost Analysis
> Compared to empirical risk minimization (ERM), TAR introduces minimal overhead. Our paper (Appendix F) includes complexity analysis, and we provide additional details here.
>
> For an $l$-layer, $d$-dimensional GCN on a graph with $n$ nodes and $m$ edges, the time cost of TAR includes:
> - **Outer Training**: $O(ld^2n + lm)$, which encompasses feature transformation and neighbor aggregation.
> - **Inner Loop**: The $k$ iterations have a time complexity of $O(kn + km)$.
>
> Real-world graphs are typically sparse, with small node degrees and larger GNN dimensions (e.g., $d=300$, max average degree $m/n=26$ in Twitch). Thus, $ld^2n \gg lm$, and since $k \ll ld^2$, TAR's additional computation is negligible. Besides, the inner loop operates in a message-passing style and is parameter-free, ensuring scalability for large-scale graphs.
>
> The table below compares TAR's runtime (epoch/s) with other methods, showing it introduces slight overhead over ERM, demonstrating **excellent scalability**.
>
> |Methods|ERM|KLDRO|GroupDRO|VREx|CIT|TAR(ours)|
> |-|-|-|-|-|-|-|
> |CBAS (4k edges)|10.1±0.2|10.2±0.7|10.3±1.2|10.8±0.2|15.0±4.3|12.0±2.2|
> |Arxiv (1.2M edges)|181.8±1.0|186.9±1.1|232.9±1.4|231.8±0.8|OOM|235.1±3.1|
>
> > Q: Why does this paper additionally consider concept shift?
>
> We would like to clarify that we follow the terminology used in GOOD benchmark, where distribution shifts are categorized into concept shift and covariate shift, with datasets specifically constructed for these cases. Our experiments strictly follow this benchmark setting, and results show that our method effectively enhances performance across both types of shifts.
>
> Thank you for your question, and we will incorporate this into our final version to avoid any misunderstandings.
>
> > Q: The bi-level optimization of min and max may lead to high training costs and instability.
>
> For TAR's training cost, we refer the reviewer to the "Training Cost Analysis" above. Regarding instability concerns, we provide additional loss visualizations at this [link](https://anonymous.4open.science/r/ICML-rebuttal-16C3). These results demonstrate that TAR achieves a smaller generalization loss without significant training instability.
>
> Should you have any additional concerns, please do not hesitate to let us know.

---

### Official Review · Reviewer_XE4R · 2025-03-24

**Overall Recommendation:** 5

**Summary:**

This paper proposes TAR, a framework which dynamically weights and reweights nodes within a Graph-Neural-Network (GNN) given the “risk”-level of nodes, incorporating topological structural information and providing robustness against shifts of distribution.

**Claims And Evidence:**

Claims are backed by theoretical proofs when made.

**Essential References Not Discussed:**

I am not aware of any essential missing references.

**Experimental Designs Or Analyses:**

The experimental design is well suited to the problem posed.

**Methods And Evaluation Criteria:**

Five classification datasets commonly used across OOD problems were used for validation, with a range of graph size and class/feature space size. Distribution shifts were defined. Baseline and state-of the art methods were compared against the newly proposed TAR framework, with TAR maximising the central tendency of test-accuracy without compromising the variance.

**Other Comments Or Suggestions:**

Graph extrapolation could have been mentioned earlier in the paper. It is used in Figure 1, but the requirements/comments are not expanded upon until section 3.3.
Page 5 Line 263 – unclear grammar used. “…which disrupts the connectivity and hinders the calculation of this Equation, we intuitively…”.
Page 13 Line 665 – Spelling error. “entorpy term”.

**Other Strengths And Weaknesses:**

This paper is concise where necessary and well-explains the TAR framework without unnecessary focus on preliminaries. In particular the backing of empirical work with theoretical proofs for understanding time complexity and limitations of the approach is commended.

**Questions For Authors:**

No questions, thank you for the paper

**Relation To Broader Scientific Literature:**

This paper builds on work in the field of OOD generalisation, in particular providing theoretical proofs alongside empirical validation for distributional robustness.

**Theoretical Claims:**

Proofs were concise and used where relevant.

---

> ### Author Rebuttal · Authors · 2025-03-31
>
> Thank you for the insights in the evaluation and the hints for revising manuscripts.
>
> > Graph extrapolation could have been mentioned earlier in the paper.
>
> Thank you for your suggestion. We will add a summary in the methodology section to introduce GE earlier.
>
> Once again, thank you for your careful review. We will thoroughly check our manuscript to avoid any grammar or spelling errors!

---

### Decision · Program_Chairs · 2025-05-01

**Decision:**

Accept (poster)

**Comment:**

The draft proposes a new alternative training proposal for Graph Neural Networks under the specific subproblem of the distribution shift between the train and test sets.
The draft gained acceptance rates by all 3 reviewers (of which at least 1 is an expert on the field and 1 is mostly agnostic). All reviewers at least confirmed that they have read the rebuttal and one of them was explicit on keeping his score.
All three reviews praised the rigorous theoretical support and some way or another also were happy with the clear and consistent evidence regarding the performance. Although on this latter I felt that inferential statistics (beyond merely descriptive statistics) was missing as the shown effect sizes are somewhat small. Among the weaknesses the contribution the reviewers appear to be more driven by "potential" issues rather than actual flaws. Among these potential issues were lack of scalability, instability, training costs and in-distribution degradation of performance among others. To all of these potential issues, the authors provided some degree of reassurance with further quantitative evidence and they did not try to duck any questions which speaks in their favor.
All in all, I can see why the reviewers agree to see this as suitable for acceptance. However given that none seem to have highlighted major actual (rather than potential) flaws, I would have expected perhaps something more solid than two 3 scores.